green chemistry/environmental chemistry

lead hydrometallurgy, replacement reaction, ultrasonic, enhancement

**Author for correspondence:**
Shiwei Li
e-mail: swli@kmust.edu.cn, lswei11@163.com, lishiweikmust@163.com

This article has been edited by the Royal Society of Chemistry, including the commissioning, peer review process and editorial aspects up to the point of acceptance.

# Ultrasonic-enhanced replacement of lead in lead hydrometallurgy process from lead leaching solution

Huimin Xie[1,2], Libo Zhang[1,2], Haoyu Li[1,2], Shiwei Li[1,2], Kaihua Chen[1,2], Bo Zhang[3] and Mi Zhou[4]

[1]State Key Laboratory of Complex Nonferrous Metal Resources Clean Utilization and [2]Faculty of Metallurgical and Energy Engineering, Kunming University of Science and Technology, Kunming, Yunnan 650093, People's Republic of China
[3]Zhengzhou Institute of Multipurpose Utilization of Mineral Resources, Chinese Academy of Geological Sciences, Zhengzhou, Henan 450006, People's Republic of China
[4]School of Metallurgy, Northeastern University, Shenyang, Liaoning 110819, People's Republic of China

HX, 0000-0002-4192-5187; SL, 0000-0001-9044-9570

In this paper, ultrasonic-enhanced replacement of lead by zinc in lead leaching solution was studied. The effects of reaction time, rotational speed, temperature, concentration of leaching solution and the ratio of the surface area of the zinc plate immersed in the leaching solution to the volume of leaching solution (S : V) were studied under both conventional and ultrasonic conditions. The optimum ultrasonic-assisted replacement conditions were as follows: the S : V of 0.04 (4 cm$^2$ 100 ml$^{-1}$), reaction temperature of 30°C, replacement time of 30 min and the concentration of leaching solution is 5 g l$^{-1}$, leading to a lead replacement rate of 94.84%. Compared with the conventional replacement process, the reaction time of ultrasonic-enhanced substitution could be reduced to one half, and the demand of reaction temperature, leaching solution concentration and other conditions were decreased accordingly. Introducing ultrasonic into the replacement reaction is promising to reduce the energy consumption in the hydrometallurgical industry also caters to the demands of environment protection.

## 1. Introduction

Lead is one of the metals which has been used by human beings since the early civilization [1], and the utilization of lead has played an important role in the progress of human civilization [2]. China has a considerable capacity for lead production in the early twenty-first century, and lead plays an increasingly vital role in the development of industry [3–6].

Currently, the production of lead by traditional smelting process is restricted due to the toxicity of lead and the difficulty of dealing with soot and gas pollutants, which cannot meet the increasing demand for lead consumption [7,8]. Therefore, the production of lead gradually shifts to the recycling of secondary resources. In those cases, more and more scholars have paid attention to the advantages of hydrometallurgical treatment of lead-containing waste residue technology, and a series of environmentally friendly hydrometallurgical treatment processes have been developed [9,10]. There are three main steps in the lead smelting process by hydrometallurgical method, which are leaching, replacement and electrolytic refining. $PbSO_4$ is initially converted into $PbCl_2$ and $CaSO_4$ (generated as slag). After that lead is replaced by zinc plate from $PbCl_2$ solution to obtain a crude lead, and finally, refined lead is obtained by electrolytic refining. The flow chart of the process is shown in figure 1. The replacement process is an important part of lead smelting by hydrometallurgical method. However, the traditional replacement of lead by zinc is insufficient because of the high consumption of zinc and the reduced activity of zinc, which results in the waste of zinc [11,12]. Therefore, the present work attempts to introduce ultrasound into the replacement process to improve the efficiency of lead smelting and reduce the energy consumption [13].

Ultrasound has been studied in many hydrometallurgy processes [14–16]. Under the action of ultrasonic waves, a series of kinetic processes occur in the tiny bubbles (holes) present in the leaching solution: oscillation, expansion, contraction and even collapse, cavitation can concentrate the sound field energy, accompanied by the bubble collapse, it is released in a very small space in the liquid, forming an instantaneous high-pressure jet of more than 1000 atmospheres and a high temperature of nearly 5000 K. When the reinforced mineral is immersed in the leaching solution, the ultrasonic wave acts on the inner and outer surfaces of the mineral granule with a strong cavitation effect, which can quickly exfoliate the surface and the inclusions, fillers and other impurities attached to the gap, thereby updating the interface. This exposes more useful mineral surface area involved in the reaction, increasing the chances of solvent molecules, ions and direct contact with useful lead minerals [17–19]. Especially in the low-grade raw material leaching process, the leaching of low-grade raw materials consumes a large amount of solvent and the leaching time is too long; therefore, the introduction of ultrasound into the leaching process of low-grade raw materials has important practical significance [20,21]. In recent years, with the development of ultrasonic technology, ultrasound is more and more widely used in the industry. Li *et al.* [22] studied a new ultrasonic treatment technology for uranium-containing wastewater with deep uranium removal. The technology can treat uranium-containing wastewater with different concentrations. It has strong adaptability, easy industrialization, high automation, good uranium removal effect and high uranium content in wastewater after 6 min of treatment. Reduction to $4.8\,\mu g\,l^{-1}$ is much better than the $50\,\mu g\,l^{-1}$ stipulated in the Regulations on Radiation Protection of Uranium Processing and Fuel Manufacturing Facilities (EJ1056-2005).

In this work, the effects of different parameters (including reaction time, rotational speed, temperature, concentration of leaching solution and the S : V) on the replacement reaction process were studied. In particular, the enhancement of ultrasound on the replacement reaction is emphatically discussed. The aim of this work is to develop an efficient and green method to improve the traditional lead hydrometallurgy process.

# 2. Material and methods

## 2.1. Materials

In this paper, the zinc leaching residue was used as the raw material. The leaching solution was obtained through the leaching process shown in figure 1. The concentration of $CaCl_2$ and the liquid–solid ratio of the leaching process was adjusted to obtain different concentrations of the leaching solution for the experimental conditions of this study. The components of raw materials were obtained by X-ray fluorescence spectrometry (XRF) as shown in table 1. We can see from table 1 that the main elemental components in the raw materials were Pb, S, Fe, Ca and Zn. The Pb content is 14.14% (where Pb content is a quantitative analysis result). The length, width and thickness of high-purity zinc plate (99.99%) used in this study were 10 cm, 1 cm and 0.1 cm, respectively. All of the chemical reagents used in this work were of analytical grade.

## 2.2. Experimental equipment and procedure

The experiment was carried out in a 78 HW-3 constant temperature magnetic stirrer (Yuhua Instrument Co., Ltd. Gongyi, China). First, the obtained leachate solution was placed in a 250 ml beaker, and the

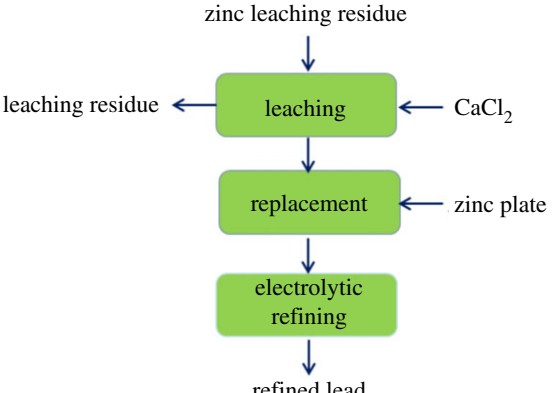

**Figure 1.** Flow chart of hydrometallurgical lead smelting.

**Table 1.** Element content of raw materials (unit %).

| Pb | S | Fe | Ca | Zn | Si |
|----|----|----|----|----|----|
| 14.14 | 13.57 | 10.32 | 8.04 | 5.72 | 5.36 |

solution in the beaker was heated to a predetermined temperature. When the solution temperature reached the set value, a zinc plate was added, and SKTC-500 ultrasonic instrument (Nanjing Ningkai Instrument Co., Ltd. Nanjing, China, frequency 20–25 kHz, output power 0–1500 W) was used to provide ultrasonic wave. The beaker was taken out and filtered with a circulating water type vacuum pump (Yuhua Instrument Co., Ltd. Gongyi, China) to obtain a filter residue and a filtrate. The lead replacement rate is calculated based on the lead content of filtrate. Five groups of single factor experiments were conducted to investigate the variation of lead replacement rate (reaction temperature, the S : V, replacement time, rotor speed and leaching solution concentration). The flow chart of the experiments is shown in figure 2, and the connection diagram of the ultrasonic replacement device used in this work is displayed in figure 3.

## 2.3. Calculation of lead replacement rate

After suction filtration, the filtrate is obtained and sent to the analytical testing centre to obtain lead content. The lead leaching rate is calculated as follows:

$$\eta_{Pb} = \frac{w_1 \times v_1 - w_2 \times v_2}{w_1 \times v_1},$$ (2.1)

where, $w_1$—lead content in leaching solution,%;

$w_2$—lead content in leaching solution after reaction, %;

$v_1$—the volume of the leach solution used in the experiment, ml;

$v_2$—the volume of leaching solution at the end of the experiment, ml.

## 3. Results and discussions

Replacement is a redox reaction in which the elementary metal that has a relatively higher standard potential is replaced by the metallic cation whose elementary metal has a lower standard potential [23,24]. It can be seen from table 2 that the potential of lead is more positive than that of zinc, so the process can happen spontaneously. The chemical reactions are shown in (3.1) and (3.2). But the traditional replacement process of lead needs to be strengthened. In this section, the replacement behaviour of lead was investigated by studying the temperature, replacement time, rotor speed and leaching solution concentration, and other parameters under conventional and ultrasonic conditions. The details are discussed in the following sections.

$$PbSO_4 + CaCl_2 = PbCl_2 + CaSO_4$$ (3.1)

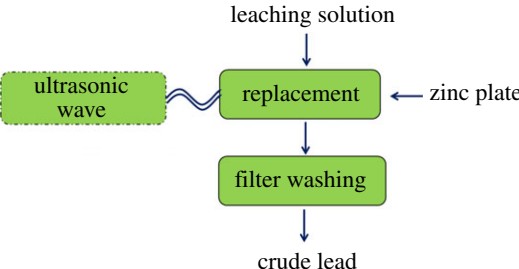

**Figure 2.** Flow chart of the experiments.

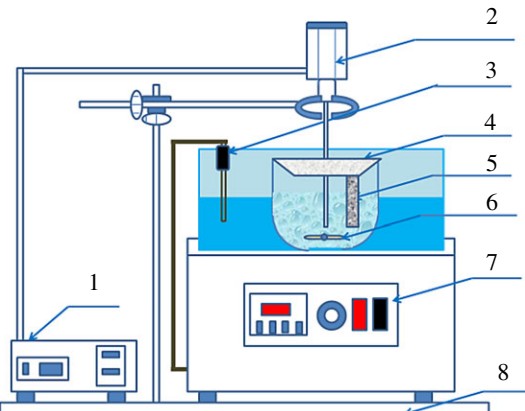

**Figure 3.** Connection diagram of the experimental device. (1—ultrasonic generator console; 2—ultrasonic radiation rod; 3—thermocouple; 4—rubber stopper; 5—zinc plate; 6—magnetic rotor; 7—water bath control panel; 8—support frame).

and

$$PbCl_2 + Zn = ZnCl_2 + Pb. \tag{3.2}$$

## 3.1. Effect of reaction time

The effect of reaction time on replacement rate was studied under ultrasonic and conventional conditions at a reaction temperature of 40°C, rotation speed of 200 rpm, $S:V = 0.04$, and leaching solution concentration of $10\,g\,l^{-1}$; the result is shown in figure 4. It can be seen from figure 4 that the replacement rate of lead under ultrasonic conditions is always higher than that of conventional experiment in the process of increasing the reaction time from 5 min to 60 min. The replacement rate of Pb can reach 97.81% after leaching for 30 min, which is even higher than that of conventional experiment of 60 min (94.32%). While the lead leaching rate did not increase significantly with leaching time extended to 30 min under ultrasonic conditions. The results show that the reaction time is significantly shortened by half, and the leaching rate of Pb can be simultaneously increased. This is because the ultrasonic wave can make the bubbles produced by the two-phase reaction break away quickly and increases the two-phase separation. The energy generated by ultrasonic vibration makes the liquid flow faster on the surface of the zinc plate [25]. Therefore, the solid–liquid contact interface is updated to enhance the replacement speed. Through the contrast experiment, the reaction time is 30 min as the experimental condition for the follow-up experiment.

## 3.2. Effect of rotor speed

Under the experimental conditions of reaction temperature of 40°C, reaction time of 30 min, $S:V = 0.04$, leaching solution concentration of $10\,g\,l^{-1}$, the effect of rotor speed was studied. The result shown in figure 5 reveals that the replacement rate of Pb increased obviously from 54.69 to 78.04% under conventional conditions when the rotational speed increased from 0 to 200 r.p.m. The replacement rate of Pb under ultrasonic experiment is always much higher than that of conventional experiment; even the replacement rate under the ultrasonic condition without agitation is higher than that of conventional replacement with the agitation of 200 r.p.m. This is mainly because the ultrasonic

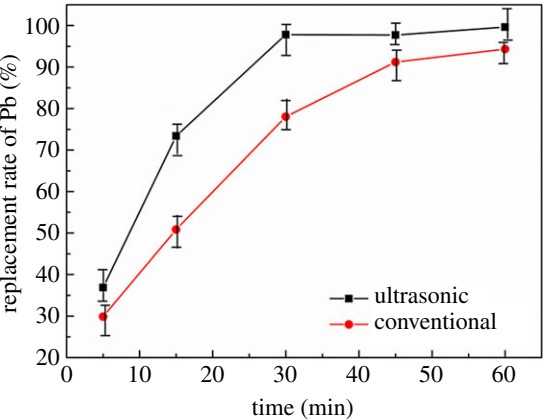

**Figure 4.** Effect of reaction time on replacement rate of lead.

**Table 2.** Electrode reaction formulae of lead and zinc and their redox potentials.

| electrode reaction | standard potential |
|---|---|
| $Zn^{2+} + 2e = Zn$ | $E^{\theta}_{Zn^{2+}/Zn} = -0.763\,V$ |
| $Pb^{2+} + 2e = Pb$ | $E^{\theta}_{Pb^{2+}/Pb} = -0.126\,V$ |

cavitation effect enhances the mass transfer effect and the activity of each substance of the reactant [26]. Therefore, the agitation has little influence on the replacement rate when ultrasonic is introduced into the process. In this case, all the subsequent experiments were carried out without agitation.

## 3.3. Effect of reaction temperature

During the hydrometallurgical leaching process, increasing the temperature can effectively reduce the bonding of mineral particles and enhance the molecular diffusion [27]. The effect of temperature on replacement rate was studied under the conditions as follows: reaction time of 30 min, $S:V = 0.04$ and leaching solution concentration of $10\,g\,l^{-1}$. It can be seen from figure 6 that the addition of ultrasonic wave can efficiently strengthen the replacement of lead by zinc plate under the same experimental conditions. The replacement rate of Pb is constantly increased with the increasing temperature under both conventional conditions and ultrasonic conditions. When the reaction temperature is fixed at 30°C, the replacement rate of lead under ultrasonic conditions can reach 94.99%, which is 49.12% higher than that of conventional experimental conditions. Moreover, the replacement rate obtained at 30°C under ultrasonic conditions is even higher than that obtained at 45°C under conventional conditions. This is because the heat transfer efficiency and the utilization rate of heat energy between two phases are very low when the temperature is low; however, the ultrasonic wave not only reduces the thermal resistance but also increases the heat transfer efficiency [28].

## 3.4. Effect of leaching solution concentration

The concentration of the leaching solution is one of the main factors influencing the leaching rate during the leaching process. To study the effect of leaching solution concentration on the replacement process, other factors are fixed at a reaction temperature of 30°C, reaction time of 30 min and $S:V = 0.04$. As can be seen from figure 7, the replacement rates of both ultrasonic and conventional process increase with the increasing solution concentration, and ultrasonic enhanced replacement leads to a higher replacement rate than that of conventional process in all solution concentrations. The replacement rate of lead under ultrasonic conditions can reach 93.91% when its concentration is only $2.5\,g\,l^{-1}$, which is 49.70% higher than the replacement rate under the conventional experimental conditions. The replacement of lead is 94.84% under ultrasonic conditions of $5\,g\,l^{-1}$, which is even higher than that obtained under conventional conditions with the solution concentration at $11.5\,g\,l^{-1}$ under conventional conditions. The introduction of ultrasonic can achieve a good replacement rate at a relatively low concentration of $PbCl_2$, in that case, the larger liquid–solid ratio can be adopted in the leaching stage when the

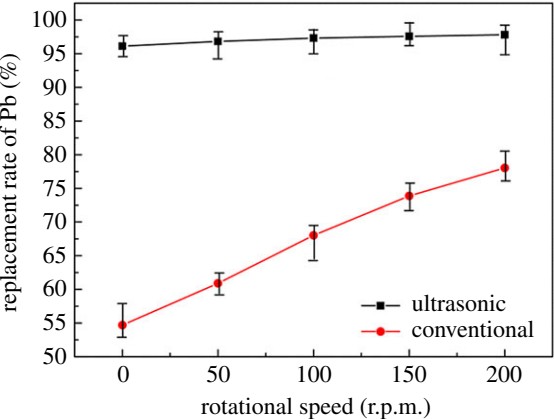

**Figure 5.** Effect of rotating speed on replacement rate of lead.

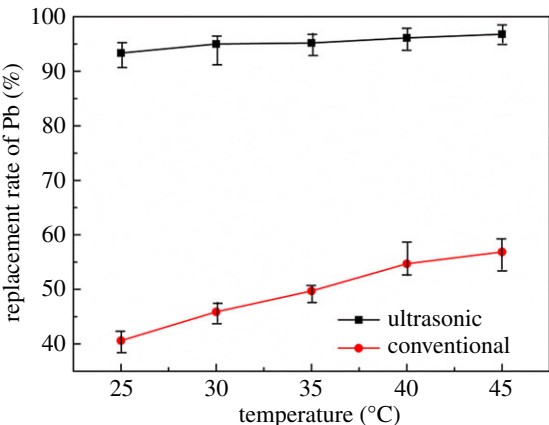

**Figure 6.** Effect of temperature on replacement rate of lead.

concentration required for the replacement process is low. Meanwhile, larger liquid–solid ratio in the leaching process can lower the viscosity of the leaching solution, increase the diffusion coefficient and increase the leaching rate in the leaching process [29]. Considering the relationship between the cost of calcium chloride and the leaching rate, the leaching solution concentration was selected as $5\,g\,l^{-1}$ as a parameter for subsequent experiments.

## 3.5. Effect of S : V

In this section, the relationship between S : V and lead leaching rate was investigated under ultrasonic treatment and conventional conditions at a reaction temperature of 30°C, reaction time of 30 min, and leaching solution concentration of $5\,g\,l^{-1}$, and the results are shown in figure 8. The replacement rate under ultrasonic conditions is always higher than that under conventional conditions in the range of S : V from 0.01 to 0.05. The replacement of Pb is 94.84% when S : V = 0.04 under ultrasonic conditions, and 47.49% higher than conventional conditions, which is even higher than that of S : V = 0.05 under conventional conditions. This is because the external field strengthening effect of the ultrasonic wave accelerates the migration rate of the molecules per unit area, and the acoustic flow action enables the lead displaced after the reaction to get out of the reaction contact surface more quickly [30]. The amount of zinc plate and the energy consumption can be reduced when the S : V is small, therefore 0.04 is elected for the best S : V.

In this study, the ultrasonic was introduced to replace lead from lead leaching solution. The high-efficiency replacement of lead was achieved, which has important research significance in the field of hydrometallurgy smelting lead.

## 3.6. Energy consumption analysis

Through the stepwise optimization process of the experimental conditions, the optimal lead replacement rate was 94.84% at a reaction time of 30 min, without rotor speed, temperature of 30°C, leaching solution

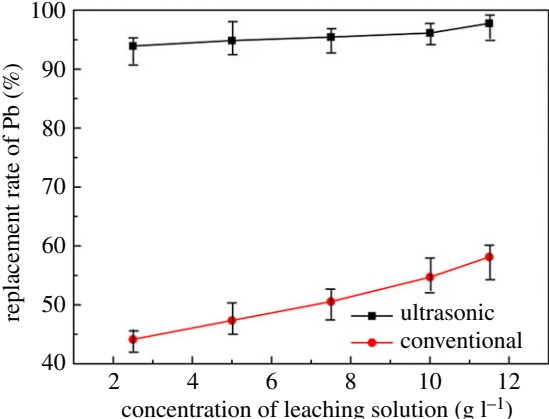

**Figure 7.** Effect of concentration of leaching solution on replacement rate of lead.

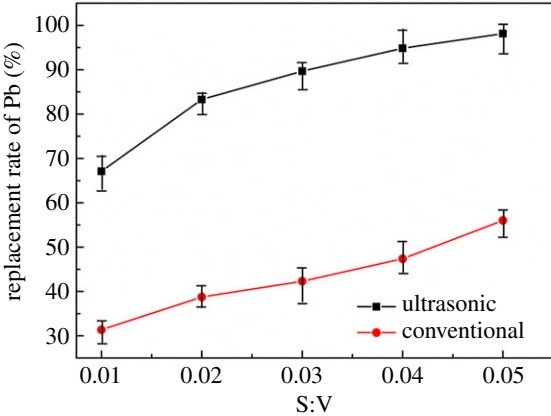

**Figure 8.** Effect of S : V on replacement rate of lead.

concentration of $5\,\mathrm{g\,l^{-1}}$ and S : V of 0.04 under ultrasonic conditions. Under conventional conditions, to obtain a similar lead replacement rate, the temperature is increased by $10°C$, the time and leaching solution concentration are increased by half, and the rotation speed needs to be 200 r.p.m. For the hydrometallurgical system, the circulation of the leaching solution reaches tens of thousands of cubic metres per day, which means about 150 000 kW h electricity will be consumed to heat up so much leaching solution at $10°C$, while a good replacement rate can be achieved at $30°C$ under ultrasonic conditions. In this case, introducing ultrasonic into the replacement process can reduce the reaction temperature and also lower the energy consumption. On the other hand, after industrial research, a $40\,\mathrm{m^2}$ solution tank requires about 40 kW of motor to stir, and the reaction time is shortened by 30 min, which saves about 20 kW h of electricity. 10 kW ultrasonic energy can be efficiently used for industrial purposes. Therefore, energy consumption can be significantly reduced with the introduction of ultrasound.

## 3.7. Scanning electron microscope analysis

Figure 9 shows the SEM images of the crude lead prepared by ultrasonic and conventional replacements and treated by suction filtration and drying. It can be seen from figure 9 that the morphology of the crude lead has changed after the interaction with ultrasonic waves: (*a*) compared with (*b*), the more massive particles are possessed, fewer fragments and the agglomeration between the lead powder particles are serious, while the coarse lead particles in the (*b*) diagram are smaller, and many layered fragments appear on the surface and the lead powder particles are dispersed more uniformly, which indicates that the agglomeration between lead powder particles can be reduced and particles can be refined under ultrasonic conditions. It is also apparent from (*c*) and (*d*) that the lead powder particles are agglomerated together during conventional replacement, while the lead powder particles are uniformly dispersed when ultrasonically displaced [31].

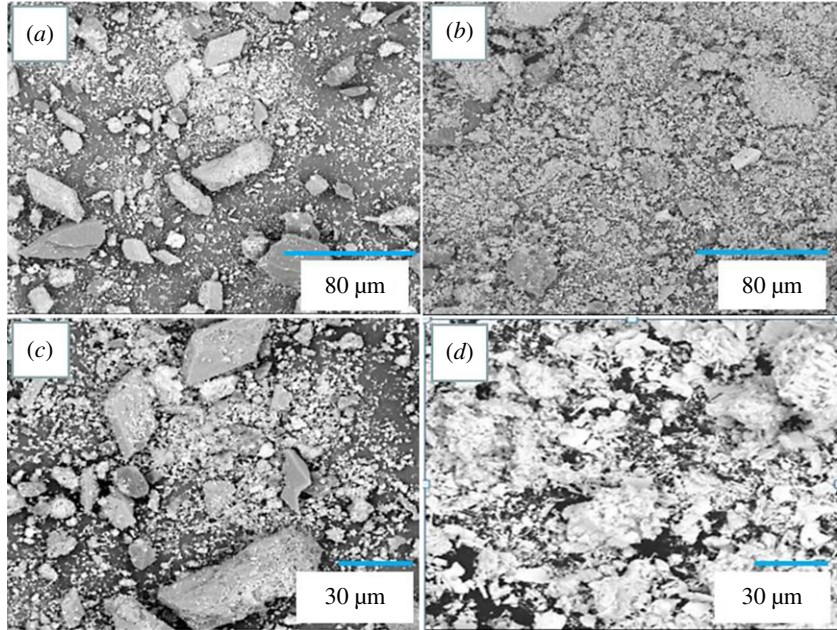

**Figure 9.** SEM images of crude lead. (*a,c*) Under conventional conditions, (*b,d*) under ultrasonic conditions.

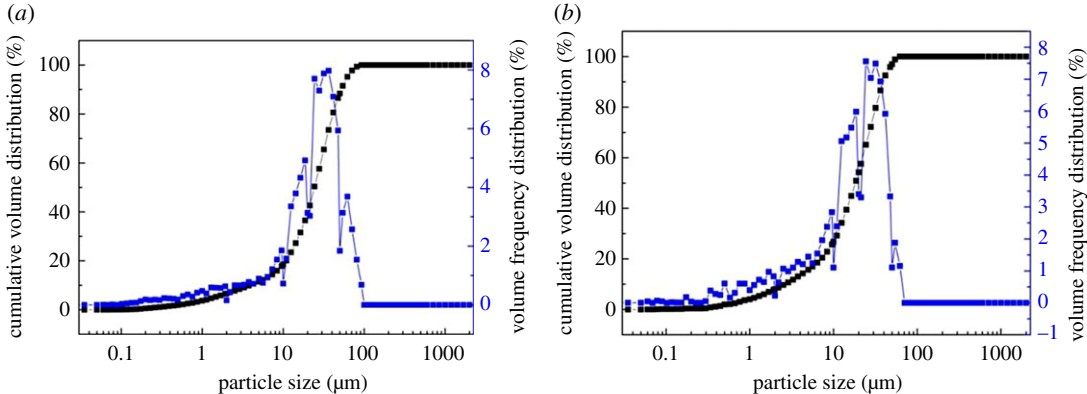

**Figure 10.** Size distribution of lead particles under conventional and ultrasonic conditions. (*a*) Under conventional conditions, (*b*) under ultrasonic conditions.

## 3.8. Particle size analysis

The results of particle size analysis of the obtained lead samples are shown in figure 10. It can be seen that the particle size is mainly distributed in the range of 10–100 µm (figure 10*a*) under conventional conditions, while figure 10*b* showed that the particle size was distributed between 10 and 65 µm under ultrasonic conditions. The lead sample under ultrasonic conditions is finer and more evenly distributed, while the lead sample under conventional conditions is obviously larger and inhomogeneous. It is indicated that the ultrasonic leaching process can actually refine the grains and ultrasonic waves can instantaneously generate stress at the interface, which in turn generates bubbles and rapidly quenches them, which acts as a peeling effect and produces a fresh surface on the surface of the zinc plate, thereby promoting the nucleation speed of the replacement process and refining the grains [32–34].

## 4. Conclusion

Ultrasonic was introduced into the process of lead replacement by zinc, and the effects of reaction temperature, reaction time, rotation speed, zinc addition amount and leaching solution concentration

on the replacement rate were discussed in detail. The following conclusions were obtained by comparing ultrasonic conditions and conventional experimental conditions:

(1) The optimal conditions for ultrasonic enhanced replacement are as follows: temperature of $30^{\circ}C$, reaction time of 30 min, no agitation, $S:V = 0.04$, concentration of $5\ g\ l^{-1}$; under these conditions, the replacement rate of lead could reach 94.84%. Compared with the conventional replacement process, the reaction time of the ultrasonic process is reduced to one half, and the requirements of all other parameters including temperature, leaching solution concentration, agitation speed and $S:V$ are decreased.

(2) Introducing ultrasonic into the replacement process could strengthen the replacement process of lead in leaching solution by zinc, refine the lead powder in the replacement process, improve the quality of lead obtained by displacement and narrow down the particle size distribution.

(3) The results of this study are expected to improve the green and efficient hydrometallurgical processing of lead.

Data accessibility. Data available from the Dryad Digital Repository: https://doi.org/10.5061/dryad.jn8108d [35].

Authors' contributions. S.L. and L.Z. designed the experiment, H.X. performed the experiment participated in the data analysis, processed the data and drafted the manuscript; H.L. carried out the statistical analyses; K.C., B.Z. and M.Z. modified the paper. All authors gave final approval for publication.

Competing interests. There are no conflicts to declare.

Funding. This work was supported by National Natural Science Foundation of China (51604135).

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
