## [Reviewer comments · Royal Society Open Science]

Review History

RSOS-190042.R0 (Original submission)

Review form: Reviewer 1

Is the manuscript scientifically sound in its present form?

No

Are the interpretations and conclusions justified by the results?

Yes

Is the language acceptable?

No

Is it clear how to access all supporting data?

Not Applicable

Do you have any ethical concerns with this paper?

No

Have you any concerns about statistical analyses in this paper?

No

Recommendation?

Accept with minor revision (please list in comments)

Comments to the Author(s)

The manuscript contains experimental results on the ultrasonically-assisted cementation of Pb(II) on Zn. The reviewer thinks that this is an interesting work and the conclusion was well-supported by data. However, it has misconceptions and some points should be better described:

The authors claim that in the leach "PbSO₄ is initially converted into soluble PbCl₂", however, they use the redox potentials of Pb²⁺ and Zn²⁺ aqua complex (Table 2) to explain the cementation process. Indeed, chloride-metal complexes must form at the studied conditions. Please include the appropriate redox pair couples. In addition, it is very important to show the reaction that describe the redox process studied (i.e. cementation).

Please verify the following reference:

[12] Li T. 2012 Experimental research on comprehensive recovery of lead, zinc and silver from a lead-silver residue. Multi. Utiliz. of Min. Res. 3, 15-20.

I tried finding above reference but it seems that it does not exist. This reference is important because justify the investigation.

Throughout the paper, there are many sentences where English grammar need to fixed.

Review form: Reviewer 2

Is the manuscript scientifically sound in its present form?

Yes

Are the interpretations and conclusions justified by the results?

Yes

Is the language acceptable?

Yes

Is it clear how to access all supporting data?

Yes

Do you have any ethical concerns with this paper?

No

Have you any concerns about statistical analyses in this paper?

No

Recommendation?

Accept with minor revision (please list in comments)

Comments to the Author(s)

This work reported the ultrasonic augmented replacement of lead by zinc in lead leaching solution, which is promising to reduce the energy consumption in hydrometallurgical industry. The results are effective and convincing, and I recommend the publication after the minor revision.

1. The error bars should be added to the figures.
2. The cost calculation on this new method should be provided that should be compared with traditional process. I am concerning the cost for electricity during the ultrasonic would be large.
3. The applicability of the ultrasonic method into a large-scale industrial application should be commented.
4. Is it possible that the high temperature during the ultrasonic would induce a dense toxic vapor of metal? This should be discussed that may have a large impact on the environment.
5. A recent publication in this field should be cited: *Angew. Chem. Int. Ed.* 2017, 56, 9331.

Decision letter (RSOS-190042.R0)

22-May-2019

Dear Dr Li:

Title: Ultrasonic-enhanced replacement of lead in lead hydrometallurgy process from lead leaching solution

Manuscript ID: RSOS-190042

Thank you for submitting the above manuscript to Royal Society Open Science. On behalf of the Editors and the Royal Society of Chemistry, I am pleased to inform you that your manuscript will be accepted for publication in Royal Society Open Science subject to minor revision in accordance with the referee suggestions. Please find the reviewers' comments at the end of this email. I apologise that this process took longer than usual.

The reviewers and handling editors have recommended publication, but also suggest some minor revisions to your manuscript. Therefore, I invite you to respond to the comments and revise your manuscript.

Please also include the following statements alongside the other end statements. As we cannot publish your manuscript without these end statements included, if you feel that a given heading is not relevant to your paper, please nevertheless include the heading and explicitly state that it is not relevant to your work. We have included a screenshot example of the end statements for reference.

- Acknowledgements

Because the schedule for publication is very tight, it is a condition of publication that you submit the revised version of your manuscript before 31-May-2019. Please note that the revision deadline will expire at 00.00am on this date. If you do not think you will be able to meet this date please let me know immediately.

Best wishes,
Dr Laura Smith
Publishing Editor, Journals

On behalf of the Subject Editor Professor Anthony Stace and the Associate Editor Professor Claire Carmalt.

RSC Associate Editor:
Comments to the Author:
(There are no comments.)

RSC Subject Editor:
Comments to the Author:
(There are no comments.)

Reviewer comments to Author:
Reviewer: 1

Comments to the Author(s)
The manuscript contains experimental results on the ultrasonically-assisted cementation of Pb(II) on Zn. The reviewer thinks that this is an interesting work and the conclusion was well-supported by data. However, it has misconceptions and some points should be better described:

The authors claim that in the leach " PbSO_4 is initially converted into soluble PbCl_2 ", however, they use the redox potentials of Pb^{2+} and Zn^{2+} aqua complex (Table 2) to explain the cementation process. Indeed, chloride-metal complexes must form at the studied conditions. Please include the appropriate redox pair couples. In addition, it is very important to show the reaction that describe the redox process studied (i.e. cementation).

Please verify the following reference:

[12] Li T. 2012 Experimental research on comprehensive recovery of lead, zinc and silver from a lead-silver residue. Multi. Utiliz. of Min. Res. 3, 15-20.

I tried finding above reference but it seems that it does not exist. This reference is important because justify the investigation.

Throughout the paper, there are many sentences where English grammar need to fixed.

Reviewer: 2

Comments to the Author(s)
This work reported the ultrasonic augmented replacement of lead by zinc in lead leaching solution, which is promising to reduce the energy consumption in hydrometallurgical industry. The results are effective and convincing, and I recommend the publication after the minor revision.

1. The error bars should be added to the figures.
2. The cost calculation on this new method should be provided that should be compared with traditional process. I am concerning the cost for electricity during the ultrasonic would be large.

3. The applicability of the ultrasonic method into a large-scale industrial application should be commented.
4. Is it possible that the high temperature during the ultrasonic would induce a dense toxic vapor of metal? This should be discussed that may have a large impact on the environment.
5. A recent publication in this field should be cited: *Angew. Chem. Int. Ed.* 2017, 56, 9331.

Author's Response to Decision Letter for (RSOS-190042.R0)

See Appendix A.

Decision letter (RSOS-190042.R1)

04-Jun-2019

Dear Dr Li:

Title: Ultrasonic-enhanced replacement of lead in lead hydrometallurgy process from lead leaching solution

Manuscript ID: RSOS-190042.R1

It is a pleasure to accept your manuscript in its current form for publication in Royal Society Open Science. The chemistry content of Royal Society Open Science is published in collaboration with the Royal Society of Chemistry.

RSC Associate Editor
Comments to the Author:
(There are no comments.)

Reviewer(s)' Comments to Author:

Appendix A

Cover letter

Dear Editor:

Thank you for your letter concerning our manuscript entitled "ID RSOS-190042 - Ultrasonic-enhanced replacement of lead in lead hydrometallurgy process from lead leaching solution ."

Those comments of you and the reviewers are very valuable and very helpful for improving our paper. We have studied comments carefully and have made correction which we hope meet with approval.

If this resubmission can meet the requirements of you and this journal now, please give us a chance and accept it. I would be greatly appreciated if you could spend some of your time on our resubmission. Looking forward to hearing from you soon.

Thank you very much for your attention to our paper.

Sincerely yours,

Shiwei Li

Address: Faculty of Metallurgical and Energy Engineering, Kunming University of Science and Technology, Kunming 650093, China

E-mail: lswei11@163.com

Statement

The revised sentences and words are indicated in red in this revised manuscript, and the statement of what has been done in this revised manuscript to each comment of the reviewers and the editor is as follow:

Reviewer 1:

Question 1: The authors claim that in the leach “ PbSO_4 is initially converted into soluble PbCl_2 ”, however, they use the redox potentials of Pb^{2+} and Zn^{2+} aqua complex (Table 2) to explain the cementation process. Indeed, chloride-metal complexes must form at the studied conditions. Please include the appropriate redox pair couples. In addition, it is very important to show the reaction that describe the redox process studied (i.e. cementation).

Response: I have revised the part and added chemical reactions of the process.

Question 2: Please verify the following reference:[12] Li T. 2012 Experimental research on comprehensive recovery of lead, zinc and silver from a lead-silver residue. Multi. Utiliz. of Min. Res. 3, 15-20.

Response: This is a Chinese document, and the author has added relevant information.

Reviewer 2:

Question 1: The error bars should be added to the figures.

Response: The single factor experimental data of this experiment is obtained by three experiments to obtain the average value. After reviewing the original data, the maximum and minimum values of all the data have been added to the original image.

Question 2: The cost calculation on this new method should be provided that should be compared with traditional process. I am concerning the cost for electricity during the ultrasonic would be large.

Response: I have added a discussion about energy consumption in 3.6.

Question 3: The applicability of the ultrasonic method into a large-scale industrial application should be commented.

Response: Ultrasonic instruments can be used in large-scale by parallel to meet industrial needs. Our team has invented a new technology for ultrasonic treatment of uranium-containing wastewater with deep uranium removal, which solves the problem of conventional ultrasonic rod corrosion and noise. This technology can handle different concentrations of uranium wastewater, easy to operate in industry, safe and stable treatment line, high degree of automation, good

uranium removal efficiency, uranium content reduced to $4.8\mu\text{g/L}$ in 6min wastewater treatment, far superior to uranium processing and fuel manufacturing facility radiation protection $50\mu\text{g/L}$ as specified in the Regulations (EJ1056-2005). As shown in the figure, the industrialization process picture shows that the installed power of the equipment is 12kW, the purification time is shortened by 60%, and the processing capacity is $15\text{m}^3/\text{h}$. The experimental process is basically the same as the principle of the uranium removal process, and the replacement process is further industrialized. Experimental principle is feasible. The author has supplemented the relevant comments in “introduction”.

Figure 1 Large ultrasonic device diagram

Question 4: Is it possible that the high temperature during the ultrasonic would induce a dense toxic vapor of metal? This should be discussed that may have a large impact on the environment.

Response: The high temperature generated by the ultrasonic wave is an instantaneous process, and the temperature is quickly lowered by the surrounding solution. The high temperature only causes the solution to instantaneously rise to about $40\text{ }^\circ\text{C}$. Although the temperature will also volatilize, the volatilization is small, and the subsequent process can be performed by the exhaust gas treatment device. So this problem can be effectively solved.

Question 5: A recent publication in this field should be cited: *Angew. Chem. Int. Ed.* 2017, 56, 9331.

Response: I have read this document, and quoted in [12]